# Genetics of Gallstones

**DOI:** 10.3390/genes16030256

**Published:** 2025-02-22

**Authors:** Agnieszka Pęczuła, Adam Czaplicki, Adam Przybyłkowski

**Affiliations:** Department of Gastroenterology and Internal Medicine, Medical University of Warsaw, ul. Banacha 1a, 02-097 Warsaw, Poland; agnieszka.peczula@uckwum.pl (A.P.); adam.czaplicki@uckwum.pl (A.C.)

**Keywords:** gallstone disease, cholelithiasis, genetic variant, ursodeoxycholic acid

## Abstract

Gallstone disease (GSD) is a common gastrointestinal disorder affecting approximately 10–20% of the global adult population, characterized by the presence of gallstones, predominantly cholesterol-based, in the gallbladder and/or biliary ducts. While many patients remain asymptomatic, more than 20% develop clinical symptoms such as abdominal pain, nausea, vomiting, jaundice, and anorexia, potentially leading to severe complications like acute cholecystitis and biliary pancreatitis. GSD has a significant genetic predisposition, with the variable prevalence of the disease according to ethnicity being highest in American and European countries and lowest in Asian and African populations. Numerous genes encoding membrane transporters involved in bile metabolism are associated with GSD, including in particular members of ATP-binding cassette transporters and others, which affect bile lithogenicity and contribute to the development of gallstones. Specific mutations in these genes are linked to an increased risk of gallstone formation, especially in individuals with certain hereditary conditions such as hemolytic diseases, thyroid disorders, and hyperparathyroidism. Advances in genetic studies have identified new variants that influence the risk of cholelithiasis, although the exact mechanisms remain partially understood in many cases. This review briefly summarizes the genetic causes of cholelithiasis, highlighting various pathogenetic mechanisms. It presents the currently used treatments and the potential implications of widely applied genetic diagnostics.

## 1. Introduction

Gallstone disease is a pathology of the digestive system present in approximately 10–20% of the world’s adult population. This complex disorder is characterized by the presence of gallstones in the gallbladder and/or biliary ducts, primarily cholesterol-based. While the majority of the patients with gallstones present no symptoms in the initial stages of the disease, more than 20% develop symptoms that include intense abdominal pain, diarrhea, fever, nausea, vomiting, jaundice, anorexia, or epigastric colic. Ultimately, GSD can result in severe complications including acute cholecystitis, acute cholangitis, and biliary pancreatitis [1]. The development of gallstones is a multifactor process influenced by various factors.

Residing in regions with limited access to healthcare or where dietary patterns favor high-fat and high-cholesterol consumption may elevate the risk of gallstone formation. Lifestyle factors play a critical role in the pathogenesis of gallstones. Diets high in saturated fats and cholesterol, coupled with insufficient fiber intake, can disrupt the balance of bile constituents, promoting gallstone formation. Furthermore, a sedentary lifestyle increases the risk of obesity and reduces gallbladder motility, both of which are significant contributors to gallstone development. Additionally, rapid weight loss—whether due to restrictive dieting or bariatric surgery—can disturb the equilibrium of bile acids and cholesterol and predispose individuals to cholelithiasis. Epidemiological studies indicate that the prevalence of gallstone disease (GSD) varies by ethnic background, with incidence rates exceeding 40% in the Americas, approximately 20% in European countries, and the lowest prevalence observed in Asian and African populations (less than 10%).

Female sex has been identified as a major risk factor for gallstone disease, with reproductive hormones playing an important role in its pathogenesis [2]. The incidence of cholelithiasis is closely linked to hormonal fluctuations during the reproductive years and may also be influenced by hormone replacement therapy [3].

Genetic predisposition is a significant determinant in gallstone disease, as evidenced by family-based and population-based case–control studies. Variants in genes encoding hepatocyte and enterocyte membrane transporters, particularly those belonging to the ATP-binding cassette (ABC) transporters family, have been most strongly associated with an increased risk of cholelithiasis. Moreover, several monogenic disorders contribute to gallstone formation, particularly those involving genetic mutations that enhance bile lithogenicity while primarily affecting organs and systems that are distinct from the liver. Notably, genetic mutations associated with familial disorders, including hyperparathyroidism, thyroid dysfunction, tubulopathies, neuromuscular dystrophies, and various hemolytic diseases, have also been implicated in gallstone pathogenesis.

In summary, gallstone formation is not attributed to a singular causative factor but rather results from the interplay of genetic predisposition, environmental factors, and lifestyle. Each of these factors can modulate bile composition and function, collectively creating conditions conducive to gallstone development. The foregoing includes genetic mutations associated with familial diseases such as hyperparathyroidism, thyroid dysfunction, tubulopathies, neuromuscular dystrophies, and others, especially hemolytic diseases. The authors aim to consolidate the latest findings on the genetic determinants of GSD, emphasizing their epidemiological associations and potential clinical applications. By integrating genetic insights with environmental and lifestyle factors, we seek to provide a perspective on gallstone pathogenesis.

## 2. *ABCG5/G8*

The genes most strongly associated with the risk of gallstone formation are the *ABCG5/ABCG8* genes encoding the transmembrane cholesterol transporter from the large group of ATP-binding cassette transporters family. Present in the membrane of hepatocytes and enterocytes, it is a key protein determining the concentration of cholesterol in bile. Several variants of this gene have been identified, which are associated with an excess of cholesterol in bile, increasing the risk of cholesterol crystallization and, consequently, the development of gallstones. These are gain-of-function variants that increase the efficiency of transmembrane cholesterol transport. In the original study by Bush et al., an association scan of SNPs in 640 individuals was performed identifying variant *D19H* (*rs11887534*) as a risk factor for gallstone disease [4]. As estimated, about 12% of the European population carries the *ABCG8 D19H* (*rs11887534*) variant and therefore is at increased risk of cholesterol gallstone formation [5]. In the last 10 years, several studies confirmed the increased risk of cholelithiasis linked to the variant (*rs11887534*), among which was a study involving 1,095 patients (529 gallstones disease cases and 566 controls) conducted by Bustos et al. [6]. Apart from the common variant (*rs11887534*), Teng et al. have found another two variants (*rs6756629*) and (*rs56132765*) associated with gallstone disease risk [7]. Another variant has been explored in the meta-analysis conducted by Joshi et al. with 8720 cases and 55,152 controls. The authors identified an increased risk of cholelithiasis in carriers of the variant *rs4245791* [8].

## 3. *ABCB4*

The adenosine triphosphate-binding cassette subfamily B member 4 (ABCB4), also known as multidrug resistance protein 3 (MDR3), is an essential transporter for the translocation of phosphatidylcholine lipids from the inner to the outer layer of the canalicular membrane of hepatocytes. Genetically determined ABCB4 deficiency or dysfunction can cause liver diseases such as phospholipid-associated cholelithiasis syndrome (LPAC), progressive familial intrahepatic cholestasis type 3 (PFIC 3), drug-induced liver injury (DILI), intrahepatic cholestasis of pregnancy (ICP), chronic cholangiopathy, adult biliary fibrosis or cirrhosis, transient neonatal cholestasis, and parenteral nutrition-associated liver disease. More than 500 mutations of *ABCB4* gene have been identified, with many missense mutations contributing to GSD [9].

LPAC is a relatively new disease entity caused by impaired secretion of phospholipids into the bile duct lumen. This disrupts the formation of mixed micelles composed of phospholipids, cholesterol, and bile acid salts. As a result, the solubility of cholesterol is reduced, promoting its precipitation into crystals, which serve as a substrate for gallstone formation. A distinctive feature of LPAC is the presence of gallstones not only in the gallbladder but also within the bile ducts, including intrahepatic bile ducts. The diagnostic criteria for LPAC include incidence of cholelithiasis before the age of 40, episodes of choledocholithiasis following a previous cholecystectomy, and a characteristic ultrasound appearance of the liver and bile tree. Its management differs from that of standard gallstone disease (GSD), as patients benefit more from ursodeoxycholic acid (UDCA) therapy rather than cholecystectomy. In a multicenter study conducted in France by Dong et al., the prevalence of LPAC in patients admitted with symptomatic gallstone disease was estimated to be between 0.5% and 1.9%. Among patients with confirmed LPAC syndrome, a mutation in the *ABCB4* gene was identified in 46% of cases [10]. The interaction between *ABCB4* variants, gender, lifestyle, and environmental factors may influence the development of clinical symptoms with varying phenotypes [11,12,13]. According to data available on the European Commission’s website, approximately 130,000 cholecystectomies were performed in France in 2021. The wider availability of genetic testing could significantly reduce the number of necessary invasive procedures by implementing oral treatment, currently based on UDCA. Spigarelli et al. identified 36 patients with LPAC, and 6 of these patients were in a group of 237 patients admitted to the hospital because of biliary symptoms. Around 17% of LPAC cases were diagnosed in patients under 40 years of age and 9.1% in patients under 50 years of age. A higher percentage of LPAC was observed in women [14]. Currently, the *ABCB4* gene is the only gene firmly associated with LPAC syndrome. To date, 158 mutations and variants only of *ABCB4* in patients with LPAC have been reported, which suggests that LPAC is a true monogenic (*ABCB4*) disease [15].

## 4. *ABCB11*

The *ABCB11* gene encodes a membrane transporter called bile salt export pump (BSEP), which actively transports conjugated bile acids from hepatocytes into the bile canaliculi. The gene was originally identified in mice under the name *Lith1*. Mutations in this gene impairing the function of the BSEP transporter are associated with increased lithogenicity of the bile and are the cause of progressive intrahepatic cholestasis type 2 (PFIC2) and primary intrahepatic stones (PIS) [5]. The precise prevalence of PFIC 2 remains undetermined. It is estimated to range between 1 in 50,000 and 1 in 100,000 live births worldwide [16]. A study involving 176 patients and 178 healthy controls of Asian ethnicity performed by Pan et al. found two variants, *rs118109635* and *rs497692,* associated with a higher risk of gallstone formation and a more severe course of PFIC2 [17].

## 5. *ABCC2*

The ABCC2 also called MRP2 (multidrug resistance-associated protein 2) is another protein of a large group of ATP-binding cassette transporters. It is expressed mainly in hepatocytes, but it can also be found in the kidneys, intestine, and placenta. The function of this protein is to transfer organic anions from the inside of the hepatocyte to the bile canaliculi under physiological conditions, mainly, conjugated bilirubin. At the same time, the MRP2 transporter is responsible for the elimination of drugs, determining the effectiveness of therapy for many diseases. Loss of function mutations of the *ABCC2* gene are responsible, among others, for the Dubin–Johnson syndrome and the formation of bilirubin stones. Currently, numerous variants of the *ABCC2* gene have been identified and are listed in the Human Gene Mutation Database. While specific variants of the *ABCC2* gene are identified in pharmacogenomics studies, distinct variants that are predisposed to the formation of gallstones are currently unknown [5,18].

## 6. *UGT1A1*

The *UGT1A1* gene encodes UDP-glucuronosyltransferase, which catalyzes the glucuronidation of bilirubin rendering it water-soluble. Mutations of the *UGT1A1* gene impairing the function of this enzyme reduce the concentration of conjugated bilirubin in bile, increasing the risk of pigment stone formation [5]. Reduced UGT1A1 enzyme activity is responsible for Gilbert’s syndrome characterized by mild intermittent hyperbilirubinemia. In the course of this syndrome, impaired metabolism of some drugs is also observed, which increases the likelihood of side effects or toxicity [19]. A study conducted by Bale et al. involving 1191 individuals with confirmed Gilbert syndrome found two variants (*rs8175347*; *rs4148323*) associated with an increased risk of GSD [20]. The combined risk of gallstones conferred by the presence of one of the pathogenic variants of the *ABCG8* or *UGT1A1* genes was estimated at 20% [21].

## 7. *SLC12A1* and *KCNJ1*

Barter syndrome is a rare renal tubular disorder characterized by sodium loss, polyuria, risk of nephrocalcinosis, and chronic renal failure. Barter syndrome type I is caused by loss-of-function mutations in sodium–potassium–chloride cotransporter gene *SLC12A1,* while type II is in potassium channel gene *KCNJ1.* In a study conducted by Puricelli et al. involving 34 patients diagnosed with Bartter’s syndrome type 1 and type 2 and 50 healthy controls, an increased risk of gallstones and persistent cholestasis was found. New missense and nonsense mutations in both *SLC12A1* and *KCNJ1* genes associated with the risk of gallstone formation at an early age have been identified [22].

## 8. *NPC1L1*

Niemann-Pick C1-Like 1 (NPC1L1) is a protein found in intestinal epithelial cells and hepatocytes. The role of NPC1L1 is to transport cholesterol from the gastrointestinal lumen to enterocytes and to reuptake cholesterol from the bile ducts to hepatocytes. This means that it plays the opposite role to the ABCG5/8 transporter. Gene variants causing reduced activity of the NPC1L1 protein led to an increased risk of gallstone formation and a reduced risk of cardiovascular diseases. These variants are *−133A>G*, *−18C>A*, *L272L* and *V1296V*, as found in a study performed by Lauridsen et al. that included 67,385 individuals of the Danish population [23]. Another study conducted by Krawczyk et al. demonstrates higher GSD incidence in patients with *rs217434* variant [21].

## 9. *CFTR*

*CFTR* gene encodes a membrane chloride channel known as cystic fibrosis transmembrane conductance regulator (CFTR). Found in epithelial cells throughout the human body it regulates the composition of many fluids such as sweat, mucus, and bile. The loss of function mutations of the *CFTR* gene leads to a more acidic pH of bile and an increased concentration of conjugated and unconjugated bilirubin in bile, which is the underlying mechanism for pigment and mixed gallstone formation. Among the numerous *CFTR* gene variants, the most common is the *F508del* variant, which accounts for 66% of disease cases. A total of 10–25% of children with cystic fibrosis and 30–72% of adults show hepatobiliary issues, in particular, “biliary sludge” and gallstones [24].

## 10. *MUC*

Genes that encode mucins are collectively referred to as MUC. Mucin proteins, produced by epithelial cells of both the bile ducts and gallbladder, are present in bile and gallstones [25].

These mucins can serve as nucleation sites for cholesterol crystals in supersaturated bile, potentially leading to gallstone formation [26]. Certain single nucleotide polymorphisms of *MUC1* and *MUC2* have been associated with gallstone formation in men but not in women in the Chinese population [27]. A 2016 Korean study reported increased expression of *MUC3* and *MUC5B* genes in patients with gallstones [28]. Additionally, the *rs3758650* variant of the mucin-like protocadherin (*MUPCDH*) gene has been identified as a genetic marker for gallstone disease [27].

## 11. Hyperparathyroidism

The exact mechanism underlying gallstone formation in hyperparathyroidism remains unclear. However, it may be attributed to reduced bile secretion, hypercalcemia, decreased motility of the sphincter of Oddi, and impaired gallbladder emptying. Genes associated with hyperparathyroidism include *MEN1*, *MEN2A*, *CDC73*, *CCND1*, *RET*, *CASR*, and *CDKN1B* [29].

In 2023, Pal et al. conducted a meta-analysis and systematic review of the prevalence of gallstone disease in primary hyperparathyroidism. Their findings indicate that patients with primary hyperparathyroidism have a 1.77-fold increased risk of developing gallstone disease [30]. The most common genetic disorders resulting in hyperparathyroidism include multiple endocrine neoplasia type 1 (Wermer’s syndrome, MEN1) and type 2A (MEN2A), hyperparathyroidism–jaw tumor (HPT-JT) syndrome, familial isolated hyperparathyroidism (FIHP). Interestingly not many studies were performed on secondary hyperparathyroidism effect on gallstone formation. One such study reported an increased risk in patients with elevated parathyroid hormone (PTH) concentration treated with peritoneal dialysis for chronic renal failure, compared to those with normal PTH levels [31]. No studies examining cholelithiasis incidence in MEN patients were found on Pubmed and MEDLINE search of “multiple endocrine neoplasm gallstone/cholelithiasis/choledocholithiasis”. However, one article on somatostatinoma stated that nearly 93% of affected patients exhibit symptoms. Yet, it did not specify which symptoms from the characteristic triad—diabetes, diarrhea, or cholelithiasis—were present or whether all three occurred simultaneously. This omission may be due to the extreme rarity of the disease, as the annual incidence of somatostatinoma is approximately 1 in 40 million [32].

## 12. Thyroid Diseases

In their 2022 article, Irina Kube and Denise Zwanziger state that while the precise mechanisms underlying the influence of thyroid function on gallstone formation remain poorly understood, thyroid hormones may affect bile flow, bile composition, and the maintenance of enterohepatic circulation [33]. Increased thyroid function can cause gallstone formation through overexpression of nuclear receptor genes participating in the metabolism of cholesterol in the liver. On the other hand, decreased thyroid function enhances the synthesis of cholesterol [34]. Notably, the correlation between thyroid hormone concentration and gallstone formation appears more pronounced in males than in females [35]. This sex-related pattern is consistently observed across multiple studies, suggesting that gender differences play a significant role in cholelithiasis.

Autoimmune polyendocrine syndromes (APS) types 2 and 3 involve autoimmune thyroid disease in conjunction with another autoimmune disorder, while multiple autoimmune syndromes (MAS) types 2 and 3 are characterized by three or more autoimmune diseases, one of which may affect the thyroid. Although APS2 is by far the most connected to human leukocyte antigen (HLA) *DR/DQ* genes (*HLA-DQ2* and *HLA-DQ8*), both APS2 and APS3 have been associated with some other susceptibility genes: cytotoxic T-lymphocyte antigen (CTLA-4) gene and protein tyrosine phosphatase non-receptor type 22 (*PTPN22*) [36,37]. However, a 2011 study of 275 Taiwanese patients found no association between symptomatic gallstone disease and *CTLA4* gene polymorphisms [38]. Similarly, a study of 500 patients, primarily from sub-Saharan Africa, with sickle cell anemia also found no correlation between single nucleotide polymorphisms in the *CTLA4* gene and cholelithiasis [39].

## 13. Muscular Dystrophy

The risk of gallstone formation in muscular dystrophies is explained by smooth muscle dysfunction. This may lead to biliary tree dyskinesis and decreased gallbladder emptying [40].

Myotonic dystrophy type 1 is caused by trinucleotide CTG repeats in the dystrophia myotonica protein kinase (*DMKP*) gene. A case study of two infants suffering from congenital myotonic dystrophy with cholelithiasis has been published: gallstones can be detected in up to 50% of patients with congenital muscular dystrophias who survived beyond the infantile period [41]. Another study on 37 patients with myotonic dystrophy type 1 stated that only 5,4% of patients had cholelithiasis [42]. This seems to be inconsistent with a study on 275 DM1 patients from Serbia, of which 36,4% developed gallstones [43].

## 14. Hemolytic Disorders

Several hemolytic diseases are associated with an increased risk of cholelithiasis. Unlike cholesterol stones, these conditions are primarily linked to the formation of pigment stones. The most common conditions include hereditary spherocytosis, sickle cell anemia, and thalassemia. β thalassemia major and sickle cell anemia patients have a higher risk of cholelithiasis compared to hereditary spherocytosis [44]. Sickle cell anemia is an autosomal recessive disorder caused by a single nucleotide polymorphism in the *HBB* (β-globin) gene. The *HLA-G* rs9380142 single nucleotide polymorphism has been identified as a genetic factor that increases the risk of cholelithiasis in patients with sickle cell anemia [39]. The introduction of gene therapy for sickle cell anemia may potentially reduce the incidence of gallstone formation [5].

Erythropoietic protoporphyria (EPP) and X-linked protoporphyria (XLP) are hereditary genetic disorders characterized by an accumulation of protoporphyrin in erythrocytes, plasma, and the biliary system. EPP results from mutations in the *FECH* gene, whereas XLP is caused by gain-of-function variants in the *ALAS2* (aminolevulinate synthase 2) gene. Both conditions are associated with an increased risk of gallstone formation [45]. Cholelithiasis may be present in 5% up to 20% of patients with EPP [46,47]. The coincidence of *UGT1A1* and *ABCG8* gene mutations increases the risk of cholelithiasis in patients with hemolytic diseases [44,48]. Noteworthy, children with gallstones should be screened for hemolytic disorders.

## 15. Treatment

Cholecystectomy remains the gold-standard treatment for symptomatic gallstone disease [49]. In cases of choledocholithiasis, endoscopic retrograde cholangiopancreatography (ERCP) is typically required. As a non-invasive alternative, extracorporeal shock wave lithotripsy may be considered for selected patients. Additionally, oral dissolution therapy with ursodeoxycholic acid (UDCA) and chenodeoxycholic acid is available for the treatment and prevention of cholesterol-based gallstones; however, the prolonged duration of therapy and high recurrence rate must be taken into account [50]. Some studies suggest that young, asymptomatic carriers of *ABCB4* mutations may benefit from UDCA therapy [5]. Peroral cholangioscopic lithotripsy (POCSL) and percutaneous transhepatic cholangioscopic lithotripsy (PTCSL) are effective treatment options for intrahepatic bile duct stones [49]. Gene therapy may represent a potential therapeutic approach potentially reducing both the incidence of symptomatic disease and the need for cholecystectomy for conditions in which gallstones arise as a secondary manifestation (like sickle cell anemia). If diagnosed correctly, progressive familial intrahepatic cholestasis (caused by *ATP8B1*, *ABCB11,* or *ABCB4* mutations) can be managed with an ileal bile acid transporter (IBAT) inhibitor or UDCA therapy [51].

## 16. Conclusions

Gallstone formation is not attributed to a singular causative factor but rather results from the interplay of genetic predisposition, environmental influences, and lifestyle. Each of these factors can modulate bile composition and function, collectively creating conditions conducive to gallstone development. Advances in genetic research have enabled the identification of numerous genes associated with gallstone disease, particularly those encoding membrane transporters involved in bile composition and cholesterol metabolism. Genes such as *ABCG5/G8*, *ABCB4*, *ABCB11*, and *UGT1A1* are crucial in increasing the lithogenicity of bile and are strongly associated with various forms of cholelithiasis, especially cholesterol-based. Additionally, other genetic disorders, such as hemolytic diseases, endocrine abnormalities (hyperparathyroidism and thyroid dysfunction), and muscular dystrophies are also associated with an increased risk of gallstone formation. Notably, differences in the genetic predisposition among ethnic groups underscore the global variability in GSD incidence. Due to the rarity of some of these conditions, in particular Barter syndrome and Myotonic dystrophy type 1, precise prevalence data remains scarce. Further research into gene–environmental interactions and the identification of new genetic variants will be important in enhancing diagnostic accuracy and developing targeted therapies for gallstone disease. While recommendations for gallstone screening with abdominal ultrasound are available for certain genetic diseases, in the vast majority, no consensus was found. The increasing availability of genetic testing facilitates the identification of patients who may require additional therapeutic interventions and those at risk of developing cholelithiasis. A standardized protocol for genetic screening of patients at a young age, with recurrent gallstone disease without other environmental or lifestyle-related factors should be established.

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
