# Peer review of "Genetics of Gallstones"

_genes, 2025, doi:10.3390/genes16030256_

Round 1

Reviewer 1 Report

Comments and Suggestions for Authors

This review summarizes concisely the genetics of cholelithiasis. The following recommendations should be addressed:

1. Avoid (or explain) gene abbreviations/symbols in the abstract.

2. ABCG5/G8: Refer to initial original studies.

3. ABCB4: Include recent prevalence data, in particular PMID: 33554096.

4. p. 4: Is ARC syndrome associated with gallstones? Provide data.

5. p. 6, Conclusions: The statement that most genetic studies suffer from limitations of small populations is not correct. The authors should look at and list all large population studies (from Scandinavia, Germany, US, UK biobank etc.).

6. p. 7: Cholestyramine should not be included as recommended medical treatment.

Author Response

Warsaw 14th February 2025

Dear Reviewer!

First and foremost, we would like to express our gratitude to the reviewers for their time and valuable feedback on our article. We highly appreciate each comment and concern raised, as our goal is to enhance the quality of our work through the peer review process.

Reviewer 1

This review summarizes concisely the genetics of cholelithiasis. The following recommendations should be addressed:

  1. Avoid (or explain) gene abbreviations/symbols in the abstract.

Response: We have deleted most of the abbreviations in the abstract and explained them as they appear in the main text.

  1. ABCG5/G8: Refer to initial original studies.

Response: Following the Reviewer’s suggestion we have added paragraphs referring to original studies [4]

  1. ABCB4: Include recent prevalence data, in particular PMID: 33554096.

Response: In the revised manuscript we have added prevalence data including suggested PMID: 33554096  [10].

  1. p. 4: Is ARC syndrome associated with gallstones? Provide data.

Response: Thank you for this valuable comment. Due to insufficient evidence supporting an association between ARC syndrome and cholelithiasis, we have excluded it from our review.

  1. p. 6, Conclusions: The statement that most genetic studies suffer from limitations of small populations is not correct. The authors should look at and list all large population studies (from Scandinavia, Germany, US, UK biobank etc.).

Response: We have removed this statement and appreciate your valuable insight.

  1. p. 7: Cholestyramine should not be included as recommended medical treatment.

Response: We have removed this misleading statement.

Again, we appreciate all of your insightful comments.

Yours sincerely

Authors

Reviewer 2 Report

Comments and Suggestions for Authors

Dear Authors,

I would like to make a few observations and suggestions for this review.
The summary is quite clear, except for the following sentences, which should be rewritten for better clarity:
"This review summarizes the genetic causes of GSD, highlighting the importance of genetic screening for early diagnosis and personalized treatment. Current therapies include ursodeoxycholic acid (UDCA), with potential for gene therapy and alternative pharmacological treatments to reduce gallstone formation and improve patient outcomes. Further research is needed to establish standardized protocols for genetic screening and treatment in populations at risk."

Introduction
The introduction should highlight that the formation of gallstones is often caused by a combination of genetic, environmental, and lifestyle-related factors. Of course, genetic factors play a key role, and the association between gallstone formation and the ABCG5 and ABCG8 genes has been demonstrated.
However, we should not forget that there are other factors and conditions with genetic determinants, such as obesity, diabetes, and certain liver diseases, which can contribute to gallstone formation.
The motivation for this review should also be emphasized, explaining why you chose to write on this topic.

ABCG5/G8
It is not enough to just cite a few studies from the literature; you need to explain the connection between these genes and gallstones. There are recent studies that describe these mechanisms, and you should expand all chapters referring to genes and gallstones. The discussions should be improved for each chapter individually.

Conclusions
Some conclusions seem disconnected and appear suddenly, as if they were not discussed earlier. Specifically, the treatment with ursodeoxycholic acid should have been addressed in the earlier chapters or in the introduction. Please either introduce it in a separate chapter or create a discussion chapter and regroup the discussions from the earlier chapters.
There are several conclusions that cite data from the literature (e.g., references 46-48, 49, 2). The conclusions should follow from this review, not merely rewrite the conclusions of other authors and studies.
We need to draw a few conclusions and future perspectives based on this review.

I look forward to the improved version.

Author Response

Dear Reviewer!

First and foremost, we would like to express our gratitude to the reviewers for their time and valuable feedback on our article. We highly appreciate each comment and concern raised, as our goal is to enhance the quality of our work through the peer review process.

Dear Authors,

I would like to make a few observations and suggestions for this review.
The summary is quite clear, except for the following sentences, which should be rewritten for better clarity:
"This review summarizes the genetic causes of GSD, highlighting the importance of genetic screening for early diagnosis and personalized treatment. Current therapies include ursodeoxycholic acid (UDCA), with potential for gene therapy and alternative pharmacological treatments to reduce gallstone formation and improve patient outcomes. Further research is needed to establish standardized protocols for genetic screening and treatment in populations at risk."

Response:  We have revised the aforementioned sentences.

Introduction
The introduction should highlight that the formation of gallstones is often caused by a combination of genetic, environmental, and lifestyle-related factors. Of course, genetic factors play a key role, and the association between gallstone formation and the ABCG5 and ABCG8 genes has been demonstrated.
However, we should not forget that there are other factors and conditions with genetic determinants, such as obesity, diabetes, and certain liver diseases, which can contribute to gallstone formation.
The motivation for this review should also be emphasized, explaining why you chose to write on this topic.

Response: We have revised the introduction and made subsections for environmental, life-style, hormonal and genetic factors, emphasizing the complex pathogenesis of gallstone disease. We have also added a part where we explained our motivation to write on this topic.

ABCG5/G8
It is not enough to just cite a few studies from the literature; you need to explain the connection between these genes and gallstones. There are recent studies that describe these mechanisms, and you should expand all chapters referring to genes and gallstones. The discussions should be improved for each chapter individually.

Response: Thank you for this valuable suggestion. In the revised manuscript we have expanded these chapters, in particular those concerning ABCG5/G8 and ABCB4 highlighting mechanisms between genes and cholelithiasis.

Conclusions
Some conclusions seem disconnected and appear suddenly, as if they were not discussed earlier. Specifically, the treatment with ursodeoxycholic acid should have been addressed in the earlier chapters or in the introduction. Please either introduce it in a separate chapter or create a discussion chapter and regroup the discussions from the earlier chapters.
There are several conclusions that cite data from the literature (e.g., references 46-48, 49, 2). The conclusions should follow from this review, not merely rewrite the conclusions of other authors and studies.
We need to draw a few conclusions and future perspectives based on this review.

Response: As suggested we have included more detailed information about the treatment which we introduced in a separate chapter.  We have thoroughly modified the conclusions hoping it improved the quality and clarity of the manuscript.

Again, we appreciate all of your insightful comments.

Yours sincerely

Authors

Round 2

Reviewer 2 Report

Comments and Suggestions for Authors

Dear authors,

Congratulations on your work!

I have no further comments regarding this review!

Warm regard